# Genome-Wide Identification and Expression Profiling Analysis of WOX Family Protein-Encoded Genes in Triticeae Species

**DOI:** 10.3390/ijms22179325

**Published:** 2021-08-28

**Authors:** Lei Shi, Ke Wang, Lipu Du, Yuxia Song, Huihui Li, Xingguo Ye

**Affiliations:** 1Institute of Crop Sciences, Chinese Academy of Agricultural Sciences, Beijing 100081, China; s5266@126.com (L.S.); wangke03@caas.cn (K.W.); dulipu@caas.cn (L.D.); 2Key Laboratory of Agricultural Biotechnology of Ningxia, Ningxia Academy of Agriculture and Forestry Sciences, Yinchuan 750002, China; songyx666@163.com; 3National Key Facility of Crop Gene Resources and Genetic Improvement, Chinese Academy of Agricultural Sciences, Beijing 100081, China

**Keywords:** Triticeae species, WUSCHEL-related homeobox, chromosomal location, phylogenetic analysis, differential expression

## Abstract

The WOX family is a group of plant-specific transcription factors which regulate plant growth and development, cell division and differentiation. From the available genome sequence databases of nine Triticeae species, 199 putative *WOX* genes were identified. Most of the identified *WOX* genes were distributed on the chromosomes of homeologous groups 1 to 5 and originated via the orthologous evolution approach. Parts of *WOX* genes in *Triticum aestivum* were confirmed by the specific PCR markers using a set of *T**riticum. durum*-*T**. aestivum* genome D substitution lines. All of these identified WOX proteins could be grouped into three clades, similar to those in rice and *Arabidopsis*. WOX family members were conserved among these Triticeae plants; all of them contained the HOX DNA-binding homeodomain, and WUS clade members contained the characteristic WUS-box motif, while only WUS and WOX9 contained the EAR motif. The RNA-seq and qPCR analysis revealed that the *TaWOX* genes had tissue-specific expression feature. From the expression patterns of *TaWOX* genes during immature embryo callus production, *TaWOX9* is likely closely related with the regulation of regeneration process in *T**. aestivum*. The findings in this study could provide a basis for evolution and functional investigation and practical application of the WOX family genes in Triticeae species.

## 1. Introduction

The Triticeae tribe belongs to the Poaceae family and is made up of more than 350 plant species in 35 genera [1]. In Triticeae plants, 111 annual species in 19 genera such as *Triticum aestivum* (bread wheat), *Hordeum vulgare* (barley), *Secale cereale* (rye), *Avena sativa* (Oat), *Triticum urartu*, *Triticum dicoccoides*, *Triticum turgidum**,* and so on, have been cultivated as crops, which provide necessary nutrition for more than two billion people in the world [1,2]. For a long time, people are interested in understanding the origination, genetic basis and evolution of Triticeae plants for their improvement. The assembling of wheat genome is a milestone in interpreting the genetic information of Triticeae plants. However, due to the large genome size of greater than 16 Gb, the genomic study on wheat has lagged behind that of rice and maize [3]. The application of modern biotechnology tools such as transgene and gene editing in plant breeding can help us to increase yield, improve quality, and enhance biotic or abiotic resistance of major crops, but the realization of these aims depends on genetic transformation. The ability of regenerating new plantlets from in vitro tissues is an obstacle that restricts the application of genetic transformation and gene editing systems [4,5].

Regeneration ability is one of many important genetically physiological traits for most plants, which enables plants to recover from wound tissues and form new organs. For modifying plants using a genetic-engineering strategy, shoot or somatic embryo production from isolated tissues or cells is an indispensable step to achieve transgenic plants. However, it is still difficult to obtain regenerated plants in the process of genetic transformation from most genotypes (especially the extensively cultivated varieties) of wheat and other Triticeae species [5,6,7,8,9]. During plant regeneration, a series of genes are expressed in an orderly manner under the regulation of auxin and cytokinin. These regeneration-related genes include *WUSCHEL-RELATED HOMEOBOX* (*WOX*), *AUXIN RESPONSE FACTOR (ARF)*, *BABY BOOM* (*BBM*), *GROWTH-REGULATING FACTOR*(*GRF*), *SCARECROW* (*SCR*), *SHORT ROOT* (*SHR*), *PLETHORA* (*PLT*), *CUP-SHAPED COTYLEDON* (*CUC*), and *YUCCA* (*YUC*) expressed during the progress of embryonic patterning, somatic embryogenesis, cell differentiation, wound reparation and epigenetic reprogramming [5,10,11,12,13,14,15]. An in-depth understanding of regeneration-related genes at the molecular level will make it possible to break through the bottleneck in genetic transformation and build a more efficient transformation system with less genotype dependence. By overexpressing a fusion protein of TaGRF4 and its cofactor GRF-INTERACTING FACTOR 1 (TaGIF1), the wheat regeneration progress was greatly speeded up with an enhanced efficiency [15]. Up-regulation of *TaTCP-1* in wheat also resulted in a higher callus regeneration rate [9]. The application of regeneration-associated genes including *WUS2* and *BBM* in crop transformation has achieved a great success, by which various maize (*Zea mays*) inbred lines and tissues, and recalcitrant genotypes of *Indica* rice (*Oryza sativa* ssp indica), sugarcane (*Saccharum officinarum*) and sorghum (*Sorghum bicolor*) were efficiently transformed for stable transgenic plants [16,17].

The WOX family is a group of plant-specific transcription factors and belongs to the homeobox (HB) transcription factor family [18]. All the identified *WOX* genes contain a conserved sequence of amino acids (60–66 residues), which is called a homeodomain (HD) encoded by the *HB* DNA sequence [19,20]. The distinctive WUS-box motif forms as T-L-X-L-F-P-X-X(T-L-[DEQP]-L-F-P-[GITVL]-[GSKNTCV]), of which the consensus structure is TLELFPLH [18]. These homolog sequences fold into a DNA-binding domain. Published data suggest that *WOX* genes act as pivotal regulators during the progress of embryonic development and polarization, plant growth and development, stem cell differentiation, embryo patterning, and flower development [21,22,23,24,25]. There are 15 *WOX* genes in *Arabidopsis thaliana*, 13 in rice, and 21 in maize [18,26,27]. In *Arabidopsis*, as a stem cell regulator, *AtWUS* is expressed in the organizing-center (OC) cells in the shoot apical meristem and regulates plant growth and shoot stem cell maintenance [28,29]. Ectopic overexpression of *WUS* genes promotes cell dedifferentiation in shoot meristem, somatic embryo formation, adventitious shoot and lateral leaf origination [29,30,31]. WUSCHEL-related homeobox proteins also protect plants from virus infection by repressing the expression of plant S-adenosyl-L-methionine–dependent methyltransferases, especially protecting plant stem cells against viral intrusion [32].

It is found that AtWOX1 possibly regulates the activity of S-adenosylmethionine decarboxylase polyamine homeostasis and/or the expression of *CLAVATA3*(*CLV3*) and has an important function in meristem development in *Arabidopsis*. Overexpression of *AtWOX1* leads to abnormal meristem development and polyamine homeostasis [33]. Normally, *AtWOX2* expresses in the zygote and early embryogenesis formation, and performs functions in correcting the apical domain development of the embryos [26]. AtWOX2 triggers the expression of *PINFORMED1* (*PIN1*), which is an auxin transport and localizes auxin to the cotyledonary tips of early embryo and root pole [21]. *AtWOX3* (*PRESSED FLOWER1, PRS1*) expresses in the peripheral layer of shoot meristem and regulates cells to form the lateral domain in vegetative and floral organs [34]. The expression of *AtWOX2* and *AtWOX3* are regulated by Leafy Cotyledon2 (LEC2), and AtWOX2 and AtWOX3 play essential roles in somatic embryogenesis [35]. *AtWOX4* is expressed in a narrow domain in cambial cells, and AtWOX4, coordinating with PHLOEM INTERCALATED WITH XYLEM (PXY), acts as a key regulator for cambium activity in the main stem [36]. *AtWOX5* is expressed in the QC of meristematic zone in root tips, regulates the columella stem cell (CSC) identity, and helps to maintain the root stem cell niche [37]. *AtWOX6* (*PRETTY FEW SEEDS2*, *PFS2*) is expressed in developing ovules and primordial and differentiating organs, regulates ovule development, and affects differentiation and maturation of leaves, outer integuments and floral primordial [38]. *AtWOX7* is expressed during all development stages of lateral root but is primarily involved in the initiation of lateral root [39]. *AtWOX8* (*STIMPY-LIKE*) and *AtWOX9* (*STIMPY*) are sister homologs [40,41] and responsible for maintaining the normal development of both basal and apical embryo lineages at the early development stage [21]. The expression of *AtWOX8* is induced by AtWRKY2 in the basal cell lineage at the initiation stage of embryogenesis [42]. *AtWOX11* plays a key role in the course of vascular cambium differentiation to new lateral root founder cells. *AtWOX11* is strongly induced and expressed in de novo root organogenesis, which is the same as its homologous *AtWOX12* [43,44]. The expression of *AtWOX11/12* is regulated by auxin, which activates the transcription of *AtWOX5/7* during the transition from root founder cells to root primordium [45]. *AtWOX13* expresses mainly in meristematic tissues to promote replum development and orchestrate fruit patterning [46]. *AtWOX14* is regulated by the CLAVATA3/ESRLIKE41/PHLOEM INTERCALATED WITH XYLEM (CLE41/PXY) pair, expressed in the procambium during stem maturation, and promotes xylem differentiation, vascular cell differentiation and lignification in inflorescence stems [47,48].

Based on the phylogenetic analysis in *Arabidopsis*, plant WOX proteins are naturally divided into three clades: WUS and WOX1 to WOX7 in the WUS clade; WOX8, 9, 11, and 12 in the intermediate clade; and WOX10, 13, and 14 in the ancient clade [18]. However, the *WOX* genes in Triticeae species have not been fully identified and characterized yet. Therefore, the objectives of this study are (1) identifying *WOX* genes in nine Triticeae species including *T**. aestivum*, *H**. vulgare*, *S**. cereale*, *A**. sativa*, *T**hinopyrum elongatum*, *T**. turgidum*, *T**. dicoccoides*, *A**egilops tauschii*, and *T**. urartu*, and aligning them onto chromosomes; (2) dividing all of the WOX proteins in these nine Triticeae species into groups by phylogenetic analysis using deduced protein sequences from all the *WOX* genes and the sequences of *OsWOX* genes from rice and *AtWOX* genes from *Arabidopsis*; and (3) analyzing the differential expression of *TaWOX* genes in five different tissues by RNA sequencing (RNA-seq) and eight tissues by quantitative real-time PCR (qPCR). Our results provide insights for further understanding the functions and evolution clarification of *WOX* family genes in Triticeae plants, and facilitate their application in gene transformation for the improvement of Triticeae plants.

## 2. Results

### 2.1. Identification of WOX Genes in Triticeae Species

In total, 43 putative *WOX* genes were obtained using the recently released IWGSC wheat genome [3], and there were still six pseudo-gene copies based on their incomplete genomic DNA sequences (Table 1). Specifically, 15 putative *WOX* genes in *H. vulgare* (Table 2), 23 putative *WOX* genes in *S. cereal**e* (Table 3), 24 putative *WOX* genes in *A. sativa* (Table 4), 14 putative *WOX* genes in *T. elongatum* (Table 5), 13 putative *WOX* genes in *A. tauschii* (Appendix A), 23 putative *WOX* genes in *T. dicoccoides* (Appendix A), 28 putative *WOX* genes in *T. turgidum* (Appendix A), and 16 putative *WOX* genes in *T. urartu* (Appendix A) were identified, respectively. Some homeologous alleles of *WOX* genes were not annotated as transcripts in the database, but were also collected and listed in the tables. For example, *TaWUSb* and *TaWUSd* were located on chromosomes 2B and 2D in *T. aestivum*, respectively (Table 1). *TdWOX12a*, *TdWOX12b*, *TdWOX7b,* and *TdWOX13b* were located on chromosomes 1A, 1B, and 3B in *T. dicoccoides*, respectively (Appendix A).

### 2.2. Identification of WUS Homoeologous Genes in Triticeae Species

In these nine Triticeae species, only one transcript of *WUS* gene was annotated as *TaWUSa* on chromosome 2A in wheat in the database (Table 1). We found the homoeologous fragments of *TaWUSa* on chromosomes 2B and 2D in *T. aestivum* (Table 1), 2H in *H. vulgare* (Table 2), 2A, 2C and 2D in *A. sativa* (Table 4), 2E in *T. elongatum* (Table 5), 2D in *A. tauschii* (Appendix A), 2A and 2B in *T. dicoccoides* and *T. turgidum* (Appendix A), 2A in *T. urartu* (Appendix A), and chromosome 2 in *S. cereale* (Table 3). According to the results of multiple sequence alignment, the full length of the open reading frame (ORF) of these homologous genes can be achieved, and their deduced amino acid sequences were highly consistent with TaWUS (Figure 1A). To understand if these genes can normally transcribe and express, promoter analysis was performed. It was shown that the promoter region of the *WUS* genes in six Triticeae plant species all contained core promoter elements, including transcription start TATA-box and AT~TATA-box, indicating they possessed potential transcriptional activity (Figure 1B). In the promoter region of *TaWUSa*, *TdWUSa*, *TtWUSa*, and *TuWUS*, a fragment of GGTCCAT existed, which is a cis-acting regulatory element involved in auxin responsiveness. Nevertheless, this element was not detected in the promoter of *AtaWUS*, *TaWUSb*, *TaWUSd*, *TdWUSb*, and *TtWUSb* (Figure 1B).

### 2.3. Chromosomal Location of WOX Genes in Triticeae Species

In general, except *ScWOX7*, *AsWOX10a*, *AsWOX10.2c*, and *AsWOX11a* genes in *S. cereale*, no *WOX* gene was found on homologous groups 6 and 7 in the genomes of these nine Triticeae plant species (Table 1 and Appendix A). In *T. aestivum*, all the *TaWOX* genes had three copies in its genomes A, B, and D. Three homologous alleles of *TaWUS* were located on chromosomes 2A, 2B, and 2D. The homologous genes of *TaWOX2* or *TaWOX12* were located on chromosomes 1A, 1B, and 1D. Three copies of *TaWOX4* or *TaWOX11* were located on chromosomes 2A, 2B, and 2D. The three homologous genes of *TaWOX7* to *TaWOX10*, *TaWOX13* and *TaWOX14* were all located on chromosomes 3A, 3B, and 3D. The three alleles of *TaWOX6* were located on chromosomes 4A, 4B, and 4D. The three alleles of *TaWOX3* or *TaWOX5* were located on chromosomes 5A, 5B, and 5D. Further investigation would be needed for the unknown chromosomal location of an incomplete transcript of *TaWOX8*. No *WOX* gene was found on homologous groups 6 and 7 in *T. aestivum* (Table 1, Figure 2A). The *HvWOX* genes in *H. vulgare* showed the similar chromosomal localization to the *TaWOX* genes in *T. aestivum* and *AtaWOX* genes in *A. tauschii*. *HvWOX2* and *HvWOX12* were located on chromosome 1H; *HvWOX4* and *HvWOX11* were located on chromosome 2H; *HvWOX7* to *HvWOX10*, *HvWOX13*, and *HvWOX14* were located on chromosome 3H; *HvWOX6* was located on chromosome 4H, and *HvWOX3* and *HvWOX5* were located on chromosome 5H. (Table 2; Figure 2B). There are additional copies of *HvWOX8* and *HvWOX10* on chromosome 3H. The *HvWOX10.1* and *HvWOX10.2* showed complete sequence consistency, but *HvWOX8.2* was shortened compared with *HvWOX8.1* (Table 2; Figure 2B).

A similar situation was observed in *A. tauschii*. *AtaWOX2* and *AtaWOX12* were located on chromosome 1D. *AtaWOX4* and *AtaWOX11* were located on chromosome 2D. *AtaWOX7* to *AtaWOX10*, *AtaWOX13*, and *AtaWOX14* were all located on chromosome 3D. *AtaWOX6* was located on chromosome 4D, *AtaWOX3* and *AtaWOX5* were located on chromosome 5D (Appendix A). Similar results were also obtained in *T. elongatum*, *T. dicoccoides**,* T. turgidum, and *T. urartu*. As expected, all the *TeWOX*, *TdWOX*, *TtWOX*, and *TuWOX* genes were located on the corresponding chromosomes of their genomes A and B because the two species only have the two genomes (Table 4 and Appendix A, Figure 2D and Appendix A). Additional copies of *TdWOX8a* and *TtWOX14a* also existed on the corresponding chromosomes.

In *S. cereale*, *ScWOX2,* and *ScWOX12* genes were located on chromosome 1R; *ScWUS*, *ScWOX4*, and two copies of *ScWOX11* were located on chromosome 2R; *ScWOX7*, two copies of *ScWOX8*, *ScWOX9*, *ScWOX10*, eight copies of *ScWOX13*, and *ScWOX14* were located on chromosome 3R; *ScWOX3* and *ScWOX5* were located on chromosome 5R; *ScWOX6* was located on chromesome 7R (Table 3, Figure 2D). All the *ScWOX* genes except *ScWOX6* were located on the corresponding chromosomes to their homoeologous chromosomes in *T. aestivum*, *H. vulgare*, *T. elongatum, T. dicoccoides*, *T. turgidum*, *A. tauschii,* and *T. urartu*.

The chromosomal location of *AsWOX* genes in *A. sativa* was complicated. The three copies of *AsWOX3*, *AsWOX4*, *AsWOX7*, and *AsWOX12* were distributed on the homoeologous chromosome groups 5, 2, 3, and 1 in order (Table 5, Appendix A). However, the three copies of *AsWOX6* were located on chromosomes 5A, 4C, and 5D; the three copies of *AsWOX8* were located on chromosomes 4A, 3C, and 3D; the three copies of *AsWOX9* were located on chromosomes 4A, 3C, and 4D; the three copies of *AsWOX11* were located on chromosomes 6A, 4C, and 5D. Moreover, only a few fragments of *WUS*, *WOX2*, *WOX5*, *WOX10* genes were found in the avaliable genome sequences of *A. sativa* (Table 5).

It is interestingly found that each *WOX* or *WUS* homoeologous gene was collinearly located on the corresponding chromosome among the nine Triticeae species. For example, *WOX2* and *12* were located on chromosome group 1 in *T. aestivum*, *H. vulgare*, *S. cereale*, *T. elongatum*, and *A. sativa*; *WOX4* and *WUS* were located on chromosome group 2 in the five species; *WOX9* was located on chromosome group 3 in the five species; *WOX3* was located on chromosome group 5 in the five species (Figure 2 and Appendix A, Table 1, Table 2, Table 3, Table 4 and Table 5 and Appendix A). The results indicated that the *WOX* or *WUS* homoeologous genes in Triticeae species were originated via orthologous evolution approach.

To verify the chromosomal locations of those WOX genes in these nine Triticeae species, partial sequences of some *Ta**WOX* genes were amplified by their specific primers using a set of *T. durum*-*T. aestivum* genome D substitution lines (Figure 3). The *TaWUSa* and its two homologs (named as *TaWUSb* and *TaWUSd*) were detected in *T. aestivum* L. cv CS (ABD genome), *T. durum* cv Langdon (AB genome), and other substitution lines except 2D(2A), indicating that the two copies *TaWUSa* and *TdWUSa* were located on chromosome 2A. *TaWUSb* was amplified in CS, Langdon, and other substitution lines except 2D(2B), indicating that *TaWUSb* was located on chromosome 2B. *TaWUSd* only appeared in CS, 2D(2A) and 2D(2B), indicating that it was located on chromosome 2D (Figure 3). Similarly, *WOX2a*, *WOX2b*, *WOX6a*, and *WOX6b* were absent in 1D(1A), 1D(1B), 6D(6A), and 6D(6B), respectively. *WOX2d* and *WOX6d* were only detected in CS and the substitution lines which contain chromosome 1D or 4D (Figure 3).

### 2.4. Evolution of WOX Family Proteins in Triticeae Species

Phylogenetic trees of WOX family proteins in Triticeae species were constructed based on the deduced protein sequences. From the phylogenetic trees, it was suggested that WOX proteins in Triticeae plants were also divided into three clades, like those in many other plant species [49,50]. However, the WOX protein classification in wheat was closer to that in rice in comparison with that in *Arabidopsis*. TaWUS, TaWOX2 to TaWOX5, TaWOX9, TaWOX13, and TaWOX14 were assigned to the same clade with the homologous proteins in rice, corresponding to *Arabidopsis* WUS clade (AtWUS and AtWOX1 to AtWOX7). TaWOX6, TaWOX7, and TaWOX10 to TaWOX12, and their homologous proteins from rice were classified into a clade, corresponding to an *Arabidopsis* intermediate clade (AtWOX8, 9, 11, and 12). TaWOX8 and OsWOX8 were clustered in separated branches, showing correspondence to an Arabidopsis ancient clade (AtWOX10, 13, and 14) (Figure 4).

HvWOX proteins were also divided into three clades: the first clade harbored HvWOX2, 3, 5, 9, 13, and 14; the second clade was for HvWOX8 only; and the third clade included HvWOX6, 7, and 10 to 12 (Appendix A). Similar to *T. aestivum**,* one branch in *S. cereale* contained ScWUS, ScWOX2 to ScWOX5, 9, 13, and 14. ScWOX6, 7, and 10 to 12 were clustered into the same branch, but ScWOX8 belonged to another branch alone (Appendix A). In *T. elongatum*, TeWUS, TeWOX2 to TeWOX5, 9, 13, and 14 were clustered in one branch, TeWOX8 was in the other branch alone, and TeWOX6, 7, and 10 to 12 were in another branch (Appendix A). In *A. sativa*, AsWOX3, 4, 9 were clustered into a branch, AsWOX6, 7, 11, 12 were in another branch, and AsWOX8 belonged to a branch alone (Appendix A). In *A. tauschii**,* one branch contained AtaWUS, AtaWOX2 to AtaWOX5, 9, 13, and 14. AtaWOX6, 7, and 10 to 12 were clustered into the same branch, but AtaWOX8 belonged to another branch alone (Appendix A). In *T. dicoccoides*, TdWUS, TdWOX2 to TdWOX5, 9, 13, and 14 were clustered in one branch, TdWOX8 was in another branch alone, and TdWOX6, 7, and 10 to 12 were in another branch (Appendix A). In *T. turgidum*, TtWOX proteins were also divided into three clades: TtWUS, TtWOX2 to TtWOX5, 9, 13, and 14 were in the first branch; TtWOX6, 7, and 10 to 12 were in the second branch; and the three copies of TtWOX8 were clustered into the same group with OsWOX8 (Appendix A). In *T. urartu*, TuWUS, TuWOX2–5, 9, 13, and 14 were grouped together, TuWOX6, 7, and 10–12 were in the same branch, and TuWOX8 belonged to a branch alone (Appendix A).

The phylogenetic tree of the WOX family proteins from nine Triticeae species was also constructed via maximum likelihood method (Figure 5). Based on the tree, it was clearly seen that the WOX proteins with the same names from these nine Triticeae species were clustered together (Figure 5), indicating that the WOX proteins were conserved in these plant species.

### 2.5. Analysis of the Conserved Motifs of WOX Proteins in Triticeae Species

All the amino acid sequences of WOX proteins in nine Triticeae species were deduced from their transcripts mentioned above. Each member contained HOX homeodomain, which were the most noteworthy symbol and defining feature of this protein family (Figure 6, Figure 7 and Appendix A). Sequences of HOX homeodomain of the three clades of WOX proteins were conserved in these nine Triticeae species (Figure 7A and Appendix A). The conserved WUS-box motif TLXLFPXX (TL-[DEQP]-LFP-[GITVL]-[GSKNTCV]) was found in TaWUS, WOX2 to WOX5, and WOX9 in these Triticeae species (Figure 6A and Figure 7B), while there was one amino acid residue change in ELXLFPXX of TaWUS and LLXLFPXX of WOX13 and WOX14 in these Triticeae species (Figure 7B). The carboxy-terminal ERF-associated amphiphilic repression (EAR) domain of L-[ED]-L-[RST]-L only exists in WUS and WOX9 (Figure 6A), and the EAR domain of WOX9 in these Triticeae species was highly conserved (Figure 7C).

### 2.6. Expression Patterns of TaWOX Genes in Various Wheat Tissues

The WOX genes were mainly expressed in the meristematic region, and played a regulatory role in the process of plant growth and tissue differentiation. We retrieved the data from expVIP website (http://wheat-expression.com, accessed date: 5 September 2020) and sketched the contours of expression pattern of *TaWOX* genes. It is shown that *TaWUS* is expressed in the root during the seedling stage, in spike during the vegetative stage, and in spike and leave/shoot during the productive stage. Its expression level was higher in spike than other organs (Appendix A). All the three are homologous of *TaWOX2* to *4*, *7*, *8*, and TaWOX*12* showed a higher expression level in developing spike than other organs, and even higher at the vegetative stage than the reproductive stage (Appendix A). The expression level of *TaWOX5* was higher in grain than that in other organs at the reproductive stage (Appendix A). *TaWOX6*, *9* to *11* showed a high transcriptional activity in root (Appendix A). The transcripts of *TaWOX10* and *TaWOX11* mainly accumulated in root at seedling stage while the expression level of *TaWOX9* was high in root at vegetative stage (Appendix A). The transcript levels of *TaWOX6b* and *TaWOX6d* in root were increased at productive stage compared with vegetative stage (Appendix A).

Furthermore, we used wheat root, stem, leave, spike at the booting stage, and anther at the heading stage as well as immature embryo and callus derived from the immature embryos at proliferative and differential stages as materials to perform expression profiling analysis of *TaWOX* genes by qPCR assay. The results indicated that expression patterns of *TaWOX* genes changed greatly in different organs at different stages (Figure 8). The expression levels of *TaWUS* and *TaWOX6* to *8* were relative high in spike (Figure 8A,B), and the expression levels of *TaWOX9* and *TaWOX11* were high in root (Figure 8B,C). Additionally, *TaWOX2* showed high activity in embryo, and *TaWOX3* and *TaWOX4* showed high expression levels in embryogenic callus and differential callus, respectively (Figure 8A).

### 2.7. Expression Patterns of TaWOX Genes during Wheat Callus Proliferation

The expression profiles of 14 *TaWOX* genes in the immature embryo and the calluses cultured for one, two, and three weeks of wheat were detected by qPCR assay. All the tested *TaWOX* genes showed a constitutive expression pattern in the calluses. In the early stage of the callus proliferation, the expression of most *TaWOX* genes was activated and up-regulated, and then repressed after two weeks (Figure 9). The expression level of *TaWUS*, *TaWOX10*, *TaWOX13*, and *TaWOX14* in the first week stage was higher than that in other two stages (Figure 9); the expression level of *TaWOX3*, *TaWOX6*, *TaWOX8*, *TaWOX9*, and *TaWOX12* was increased after the callus initiation and constitutively expressed in the calluses (Figure 9); the expression level of *TaWOX2*, *TaWOX7*, and *TaWOX11* was always low during the callus production period (Figure 9); meanwhile, the expression level of *TaWOX4* and *TaWOX5* was down-regulated during the callus proliferation course (Figure 9A). Particularly, TaWOX9 was highly expressed in the calluses of Fielder (Figure 9B).

## 3. Discussion

In Triticeae species, wheat and barley are two important crops globally which account for a large proportion of food production in the world. With the completion of wheat genome assembly and annotation, a great progress on functional genomic study in Triticeae plants, especially in wheat, has been achieved [51,52,53,54]. It is well-known that the wheat genome was originated from the natural hybridization of its three ancestor species. Therefore, the wheat genome consisting of three genomes of A, B, and D has a large number of repeated gene sequences, and most wheat genes have three or more copies [55]. In the present study, we identified 43 putative *WOX* gene copies in the genome of *T. aestivum*, 42 of which were consistent with the result reported by Li et al. [56], and a new locus of *TaWOX8* was added to the results of TaWOX family. Particularly, we firstly identified 17 putative *WOX* genes in *H. vulgare*, 23 putative *WOX* genes in *S. cereale*, 24 putative *WOX* genes in *A. sativa*, 14 putative *WOX* genes in *T. elongatum*, 13 in *A. tauschii*, 30 in *T. turgidum*, 25 in *T. dicoccoides,* and 16 in *T.*
*urartu*. There were still several duplicated copies of the *WOX* gene such as *TaWOX14a, TaWOX14d, HvWOX10, TdWOX14**, ScWOX11,* and*ScWOX13*. A few *WOX-*like pseudo genes were found to be scattered over Triticeae genomes, which might be a duplication of *WOX* genes or the other genes losing transcriptional activity during their evolution progress.

WUS plays an indispensable role on stem cell niche maintenance in shoot apical meristem (SAM), lateral primordia differentiation and other diverse cellular processes [29]. The deficiency of the *WUS* gene will lead to the loss of function of SAM and terminated plant growth [28]. However, only the allele of *TaWUS* located on chromosome 2A was annotated as a transcript. *TdWOX12a*, *TdWOX12b*, *TdWOX7b,* and *TdWOX13b*, which have a high sequence identity with their homologous genes from wheat, were also not annotated as transcripts in the database. The DNA sequences and deduced protein sequences of four genes *TdWOX12a*, *TdWOX12b*, *TdWOX7b*, and *TdWOX13b* were added into the WOX members in the six Triticeae species (Appendix A). In barley, the annotation of *HORVU1Hr1G087940* and *HORVU1Hr1G087950* and their deduced protein sequences A0A287GM87 and A0A287GM65 are actually originated from *HvWOX12* (Table 2).

In previous studies, the classification and naming of *WOX* genes in wheat were confusing to some extent. This might be attributed to the different naming scheme of *WOX* genes in *Arabidopsis* and rice [18,26,27]. For example, the *TaWOX5* reported by Zhao et al. [57] was regarded as *TaWOX9* due to its high similarity to *OsWOX9,* even though it showed a close similarity to *AtWOX5* in all the WOX members in *Arabidopsis* (Figure 4). Several reported *TaWOX* members such as *TraesCS3A02G358100*, *TraesCS3B02G391100*, *TraesCS3D02G352500*, *TraesCS3A02G358200*, *TraesCS3A02G358400*, *TraesCS3B02G391200, TraesCS3D02G352600*, and *TraesCS3D02G352700* on chromosomes 3A, 3B, and 3D, respectively, were named *TaWOX13* and *TaWOX14* [51] according to new nomination regulations. However, *TaWOX13* was not similar to *AtWOX13* or *OsWOX13*, and *TaWOX14* was also not similar to *AtWOX14* in transcripts*,* while *TaWOX13* and *TaWOX14* were similar to the homologs of *TaWOX5* according to phylogenic analysis (Figure 4). The WOX13 and WOX14 in other Triticeae species showed the similar phylogenetic relationship with WOX5 members (Figure 5).

All the *TaWOX* genes in wheat have three or more copies. Due to their sequence similarity, it is difficult to distinguish the expression level of each copy of *TaWOX* genes. A feasible approach was applied to estimate the amount of mRNA by calculating transcript amount of each copy. Zhao et al. indicated that the transcriptional level of the individual *TaWOX5* allele was varied during the period of callus growth in wheat [57]. Based on the results in the present investigation, the expression profiles of other *WOX* alleles were also changed in different wheat organs, which need to be justified by further research.

In the plant regeneration system through somatic embryogenesis from somatic tissues, the activation of an embryonic developmental program is an essential step [58], which is widely adopted in plant genetic engineering. As a group of important transcription factors, WOX family proteins are involved in fate transition of stem cells, cell growth regulation, and somatic embryogenesis process. Overexpression of *WOX* genes or co-expression of *WOX* genes and other regeneration-related genes is a potential strategy to resolve the difficult transformation situation in many plant species. It was demonstrated that the overexpression of *ZmBBM* and *ZmWUS2* produced high transformation frequencies in maize, sorghum, sugarcane, and indica rice [16]. A recent study revealed that the overexpression of *ZmBBM* and *ZmWUS2* was driven by the specific promoters of *ZmPLTP* and *ZmAxig1* to embryo development, respectively, initiated abundant somatic embryos, and further developed into plantlet directly without a callus phase [17]. Besides the *WUS* genes, other *WOX* gene members were also proved to play a part role in the regeneration process. The *AtWOX5* regulated wound healing after laser ablation and root tip excision, in which the molecular program of callus induction and development is similar to those in root establishment, even if the callus is derived from the non-root tissue [59,60]. Following an auxin maximum after wounding in leaf explants, WOX11 and 12 which are involved in the specification of a hypophysis-like root founder cell activate *WOX5* and *7* and promote the root primordia initiation [44,45].

Based on the findings achieved in this investigation, *TaWUS*, *TaWOX8*, *TaWOX10*, and *TaWOX14* were greatly up-expressed in the differentiated callus (Figure 8A–C), which indicated that these genes may be related to callus differentiation process. In the callus proliferation process of the immature embryos of four wheat cultivars used in this study, the expression of the 14 *TaWOX* genes showed a similar pattern (Figure 9). However, in the model spring wheat genotype Fielder with high regeneration and transformation efficiency [7,61], *TaWOX9*, *TaWOX6*, *TaWOX10*, and *TaWOX12* all had a significant up-regulation during callus production process (Figure 9). Considering that *TaWOX9* was a homologous gene of *AtWOX5/AtWOX7*, and *TaWOX6*, *TaWOX10,* and *TaWOX12* showed a high homology with *AtWOX11/AtWOX12* (Figure 4), we speculate that the expression pattern of these *TaWOX* genes might be related to the regulation of regeneration process in *T. aestivum*.

## 4. Materials and Methods

### 4.1. Materials and Cultivation Conditions

Wheat cultivars Fielder, CB037, Chinese Spring (CS), and Ningchun4 were used as plant materials to conduct gene identification and expression analyses. A set of *T. durum-T. aestivum* genome D substitution lines and their genetic background Langdon (LD), which were kindly provided by Dr. Steven Xu at the Northern Plains Crop Science Laboratory of the USDA-ARS, North Dakota, USA, and genetically identified by Prof. Zhishan Lin at the institute of Crop Sciences (ICS), Chinese Academy of Agricultural Sciences (CAAS), Beijing, China, were used to verify the chromosomal localization of the *WOX* genes identified in this study. In each of these disomic substitution lines, a pair of A-genome or B-genome chromosome in the tetraploid wheat *T. durum* was replaced by a corresponding pair of D-genome chromosomes from *T. aestivum.* For example, in substitution line 1D(1A) chromosome 1D from *T. aestivum* D replaces the chromosome 1A in *T. durum*. Thirty seeds of those wheat materials were planted as a trail with 1 m in length and 20 cm in width in the experimental station of ICS, CAAS, Beijing, China, under natural soil conditions without stress.

### 4.2. Database Used for Searching WOX Family Genes in Triticeae Plants

Twenty-six predicted WOX family protein sequences of *T. aestivum* were obtained from Plant TFDB database (http://planttfdb.gao-lab.org/, accessed date: 1 September 2020) and retrieved Genbank (https://www.ncbi.nlm.nih.gov/genbank, accessed date: 1 September 2020) with AtWOX of *Arabidopsis* and OsWOX of rice (Appendix A). Using all of the protein sequences above as queries to conduct a TBLASTN search on Gramene (http://ensembl.gramene.org/Tools/Blast, accessed date: 2 September 2020) and URGI (https://urgi.versailles.inra.fr accessed date: 2 September 2020) for the identification of WOX protein- encoded genes in the genome of *T. aestivum*. Then, a BLASTN search with sequences of *TaWOX* genes was performed in the genomes of *H. vulgare*, *T. dicoccoides*, *T. turgidum*, and *A. tauschii.* Using all the sequences of *TaWOX* genes as queries to conduct a TBLASTN search on GrainGenes (https://wheat.pw.usda.gov, accessed date: 29 July 2021) for the identification of WOX protein-encoded genes in the genomes of *A. sativa* and *T. urartu*, a TBLASTN search on WheatOmics (https://http://202.194.139.32/, accessed date: 29 July 2021) for the identification of WOX protein-encoded genes in the genomes of *S. cereale* and *T. elongatum* All the genetic analysis was carried out using these protein sequences of these nine Triticeae plants. Based on the BLAST results from Gramene and URGI, the *WOX* genes from the Triticeae plants were located on exact chromosomes. The location chart was created by MapGene2Chrom web v2.1 (http://mg2c.iask.in/mg2c_v2.1/ accessed date: 29 July 2021).

### 4.3. Phylogenetic Trees Construction

The full-length of the WOX proteins of Triticeae species were aligned by ClustalW algorithm. Phylogenetic analysis and phylogenetic tree construction were performed by the MEGA X program (version 10.0.5) [62] using the Maximum Likelihood method, JTT matrix-based model [63], and 1000 bootstrap replicates. The initial tree for the heuristic search was obtained automatically by employing Neighbor-Join and BioNJ algorithms to a matrix of pairwise distance estimated using a JTT model, and then selecting the topology with a superior log likelihood value. In total, 130 amino acid sequences were used for this analysis. All the positions with less than 95% site coverage were eliminated, and alignment gaps fewer than 5%, missing data, and ambiguous bases were allowed at any position (partial deletion option). Consequently, 93 positions were remained in the final dataset. Sequences of TaWOX proteins were aligned with AtWOX and OsWOX proteins, and a phylogenetic tree was constructed to confirm classification and phylogenetic relationship of the identified TaWOX members. Then, taking OsWOX proteins as model, the phylogenetic trees were constructed between OsWOX and HvWOX, OsWOX and TdWOX, OsWOX and TtWOX, and OsWOX and AtaWOX members to name and classify the WOX members in the six Triticeae species.

### 4.4. Conserved Protein Motif Analysis

The conserved domain HD was identified by SMART software (http://smart.embl-heidelberg.de/, accessed date: 29 July 2021). The distinctive WUS-box motif as TLXLFPXX(T-L-[DEQP]-L-F-P-[GITVL]-[GSKNTCV]) and the EAR domain as LXLXL(L-[ED]-L-[RST]-L) were both defined in a strict sense. TEXshade software [64] was employed to perform the multiple sequence alignments for HD domains, WUS-box motifs, and EAR motifs. The logo diagrams were drawn by canonical conserved residues including HD domains, WUS-box motifs, and EAR motifs by SeqLOGO in TBTools (version 1.075) [65].

### 4.5. DNA Isolation and PCR Analysis

Wheat genomic DNA was isolated by NuClean Plant Genomic DNA kit (Cwbio, CW0531M, Beijing, China) from the leaf samples at the three-leaf stage. The PCR reaction system (20 μL) contained 10 μL of 2× Taq Master Mix (containing Mg^2+^ and dNTP, Vazyme), 0.5 μL of each forward primer and reverse primer (10 mM), and 1 μL of gDNA (1 μg·μL^−1^), adding ddH_2_O up to 20 °C. Sequences of all the primers used for the detection are shown in Appendix A. The thermal cycling conditions were 94 °C for 5 min, 35 cycles of 94 °C for 20 s, 60 °C for 20 s, 72 °C for 30 s, and then 72 °C for 10 min.

### 4.6. Callus Induction from Wheat Immature Embryos

The immature seeds were collected from wheat cultivars Fielder, CB037, CS, and Ningchun4 15 days post anthesis (DPA), and their immature embryos were isolated after surface sterilization as described previously [7,61]. The scutella with the flat side up after removing the embryonic axes were inoculated onto an MS medium containing 2,4-D 2.0 mg L^−1^ and cultured at 25 °C in darkness for 14 days. Then, the primary calluses were sliced vertically into halves and cultured on the same medium under the same conditions for another 14 days. Then, the embryonic calluses were cultured for differentiation on 1/2 MS medium in a photoperiod of 14 h light and 10 h darkness [7,61]. Callus samples were collected when they were cultured on callus induction medium for one to three weeks, and on calluses differentiation medium for one week.

### 4.7. RNA Isolation and qPCR Analysis

Wheat samples were collected from the seedlings at the three-leaf stage for roots, stems, and leaves, from the adult plants at the booting stage for young spikes, and at the heading stage for anthers. Wheat immature embryo samples were collected at 15 DPA, and callus samples were collected after culturing for one week, followed by two or three weeks on callus production medium, and one week on differentiation medium.

Total RNA was extracted using TransZol Up Plus RNA Kit (Transgen, ER501-01, Beijing, China), and a reverse transcription reaction was performed using the PrimeScript™ RT reagent (Takara, Dalian, China) according to the manufacturer’s protocol. The qPCR was performed on a ABI7500 Thermal Cycler using 2× RealStar Green Fast Mixture (with ROX II, Genestar, Beijing, China). The *TaActin* (Genbank: AB181991) was used as an internal control, and three biological replicates were adopted. Gene-specific primers were designed with Primer Premier (Version 6.00) software (Appendix A). Each qPCR reaction system (20 μL) contained 10 μL of 2× RealStar Green Fast Mixture, 0.4 μL of forward primer (10 mM), 0.4 μL of reverse primer (10 mM), and 1 μL of diluted cDNA (200 ng·μL^−1^). The thermal cycling conditions included 95 °C for 5 min, 40 cycles of amplification (95 °C for 15 s, 60 °C for 15 s, and 72 °C for 30 s), and 95 °C for 10 s at dissociation stage, followed by 65–95 °C with increments of 0.5 °C for 0.05s.

### 4.8. Expression Analysis of TaWOX Genes Using RNA-seq Data

RNA-seq data of 43 *TaWOX* genes was downloaded from expVIP (http://wheat-expression.com/ accessed date: 5 September 2020). Their expression levels in roots and leaves/shoots at the seedling stage, spikes at the vegetative stage, and grains at the reproductive stage were analyzed.

### 4.9. Statistical Analysis

The SPSS 19.0 software package was employed to statistically analyze the expression data of the target genes achieved by qPCR. Statistical comparisons of multiple sets of data was carried out by Duncan’s multiple range test. The histogram was made using the Excel software.

## 5. Conclusions

To our knowledge, this is the first study on genome-wide and contrastive analysis on *WOX* family genes in annual Triticeae plant species. In total, 199 *WOX* genes were identified, including 43 in *T. aestivum*, 28 in *T. turgidum*, 23 in *T. dicoccoides*, 23 in *S. cereal**e*, 24 in *A. sativa*, 14 in *T. elongatum*, 15 in *H. vulgare*, 13 in *A. tauschii*, and 16 in *T. urartu*. The homoeologous genes of *TaWUSb*, *TaWUSd*, and *WUS* in the other five Triticeae species were annotated, which were predicted to express normally according to the promoter element analysis. Four novel homologous alleles of *TaWOX* genes including *TdWOX12a*, *TdWOX12b*, *TdWOX7b,* and *TdWOX13b* were also identified in *T. dicoccoides*. All of these WOX members showed similar chromosomal location arrangement and a collinearly orthologous evolution relationship in Triticeae species. Based on the RNA-seq data in the wheat-expression database and qPCR array results, *TaWOX* genes were found to have a tissue-specific expression feature, and the expression of part-*TaWOX* genes were closely associated with the regulation of the regeneration process in *T. aestivum*. The results obtained in this study would be helpful to further understand the molecular function and evolutionary relationship of *WOX* family genes in Triticeae plants, and potentially apply them in plant genetic transformation in the future.

## Figures and Tables

**Figure 1 ijms-22-09325-f001:**
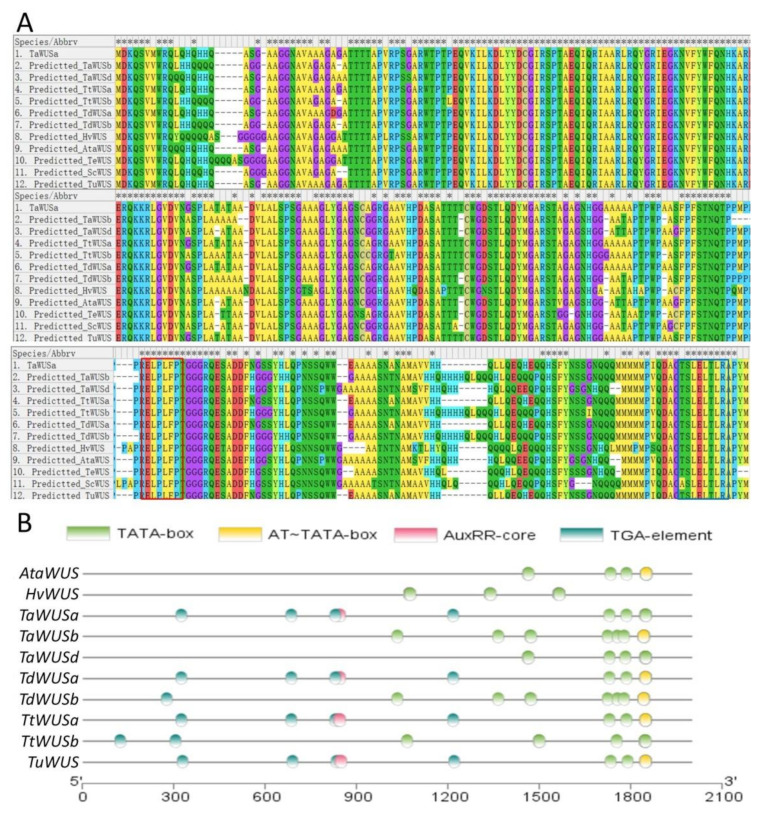
Multiple-sequence alignment and element prediction of the promoters among *TaWUS* and other predicted *WUS* genes from nine Triticeae species. (**A**) Multiple sequence alignment among TaWUS and other predicted WUS proteins. Alignment of protein sequences was conducted by ClustalW algorithm using MEGA X. The position of conserved WUS-box motif was shown in red box, and the position of EAR domain was shown in blue box. The conserved amino acid sites were marked with asterisk (*). (**B**) Element prediction of the promoter regions of *TaWUS* and other predicted *WUS* genes in six Triticeae species. TATA-BOX elements and their positions in the promoters were displayed in green oval, AT~TATA-BOX elements and their positions in yellow oval, cis-acting regulatory elements involved in auxin responsiveness AuxRR-core and their positions in red oval, and auxin-responsive TGA-elements and their positions in blue oval.

**Figure 2 ijms-22-09325-f002:**
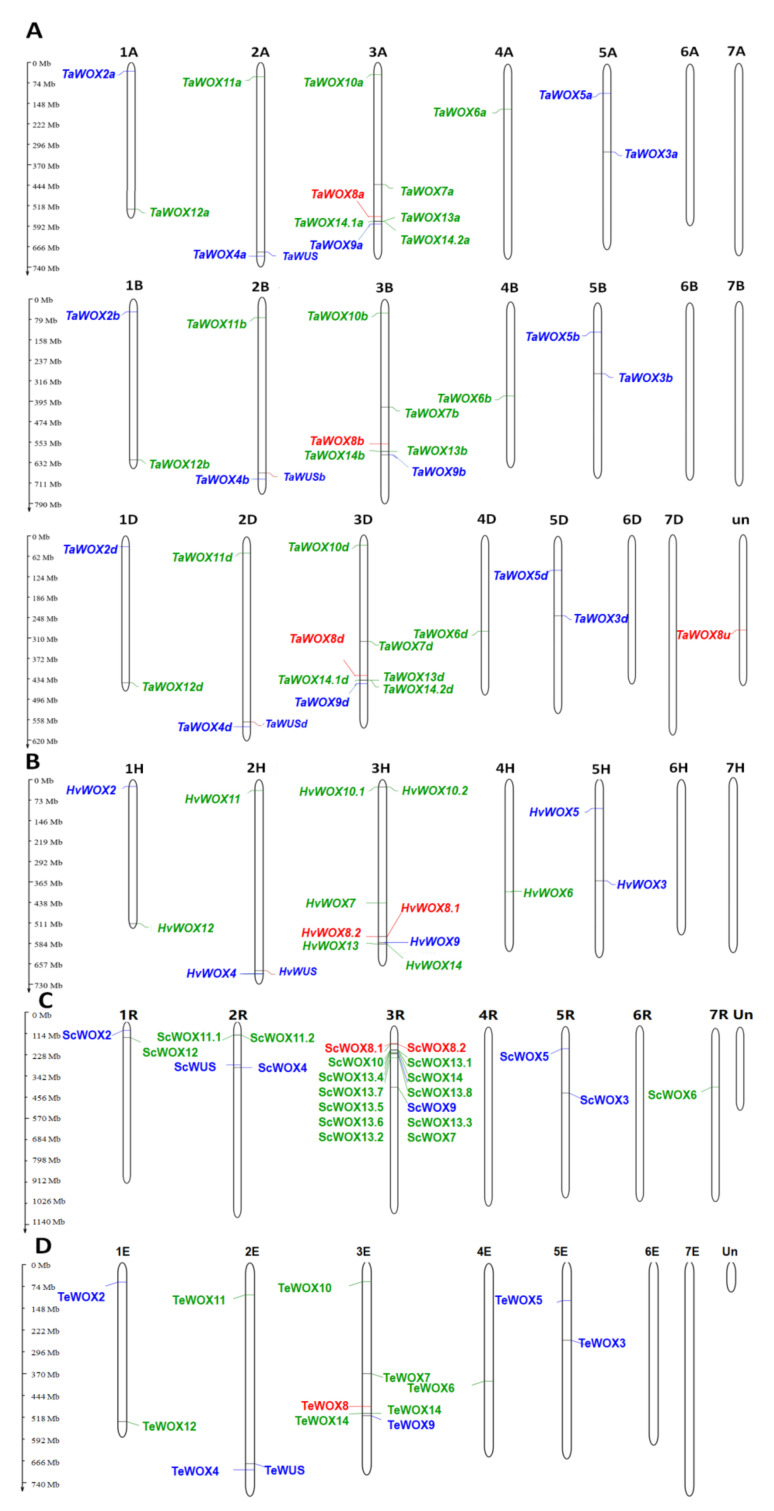
Chromosomal locations of *WOX* genes in *T. aestivum* (**A**), *H. vulgare* (**B**), *S. cereale* (**C**) and *T. elongatum* (**D**). The number of chromosomes was labeled on the top of each chromosome. The location of each *WOX* genes was marked on the chromosome. The WOX members in WUS clade, intermediate clade and ancient clade were shown as blue, green, and red types, respectively.

**Figure 3 ijms-22-09325-f003:**
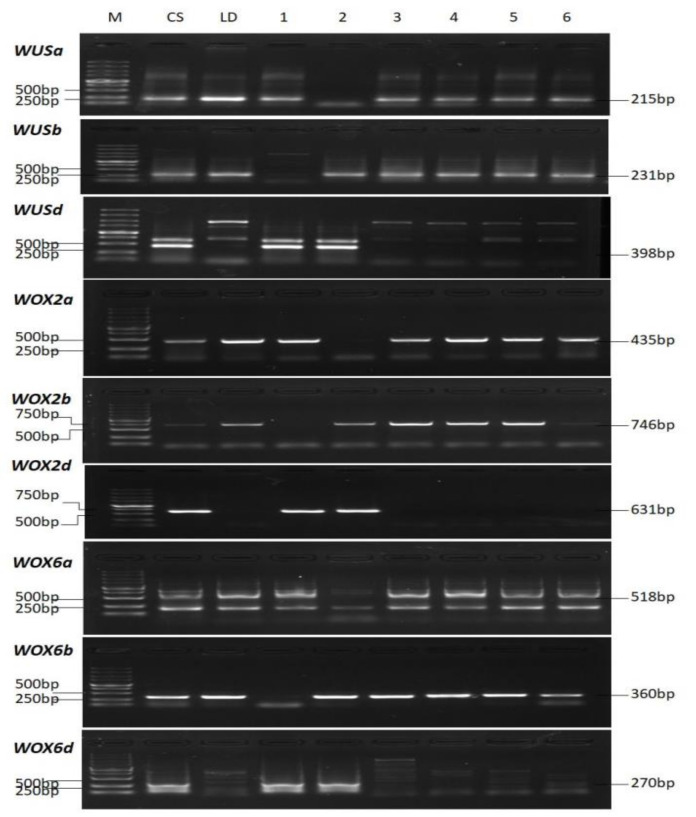
Verification of chromosomal locations of several *TaWOX* alleles by specific PCR amplification. Gel electrophoresis of specific fragments of *TaWUS*, *TaWOX2* and *TaWOX6* alleles. M, DNA molecular marker; CS, PCR product of Chinese Spring (*T. aestivum*); LD, PCR product of Langdon (*T. durum*)*;* samples 1–6 in row *WUSa* were: substitution lines 2D(2B), 2D(2A), 1D(1A), 3D(3A), 4D(4A), and 5D(5A); samples 1–6 in row *WUSb* were: 2D(2B), 2D(2A), 1D(1B), 3D(3B), 4D(4B), and 5D(5B); samples 1–6 in row *WUSd* were: 2D(2B), 2D(2A), 1D(1A) and 1D(1B), 3D(3A) and 3D(3B), 4D(4A) and 4D(4B), and 5D(5A) and 5D(5B); samples 1–6 in row *WOX2a* were substitution lines 1D(1B), 1D(1A), 2D(2A), 3D(3A), 4D(4A), and 5D(5A); samples 1–6 in row *WOX2b* were 1D(1B), 1D(1A), 2D(2B), 3D(3B), 4D(4B), and 5D(5B); samples 1–6 in row *WOX2d* were 1D(1B), 1D(1A), 2D(2A) and 2D(2B), 3D(3A) and 3D(3B), 4D(4A) and 4D(4B), and 5D(5A) and 5D(5B); samples 1–6 in row *WOX6a* were substitution lines 4D(4B), 4D(4A), 1D(1A), 2D(2A), 3D(3A), and 5D(5A); samples 1–6 in row *WOX6b* were 4D(4B), 4D(4A), 1D(1B), 2D(2B), 3D(3B), and 5D(5B); samples 1–6 in row *WOX6d* were: 4D(4B), 4D(4A), 1D(1A) and 1D(1B), 2D(2A) and 2D(2B), 3D(3A) and 3D(3B), and 5D(5A) and 5D(5B).

**Figure 4 ijms-22-09325-f004:**
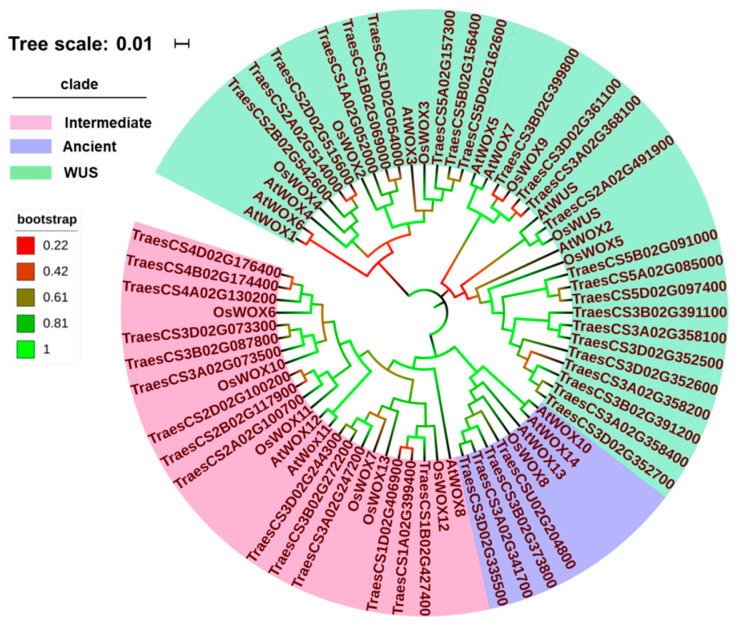
Phylogenetic relationships among WOX proteins in *T. aestivum*, rice and *Arabidopsis.* Phylogenetic tree was constructed based on the sequences of WOX proteins in *T. aestivum*, rice and *Arabidopsis,* performed by the MEGA X using neighbor-joining approach with 1000 bootstrap replicates. WUS clade harbors WUS, WOX2 to WOX5, WOX9 in *T. aestivum* and rice, TaWOX13, TaWOX14, AtWUS and AtWOX1 to AtWOX7; intermediate clade contains WOX6, WOX7, and WOX10 to WOX12 in *T. aestivum* and rice, AtWOX8, 9, 11, and 12; and ancient clade contains WOX8 in *T. aestivum* and rice, AtWOX10, 13, and 14. Scale plate and legend in upper left displayed tree scale and bootstrap value.

**Figure 5 ijms-22-09325-f005:**
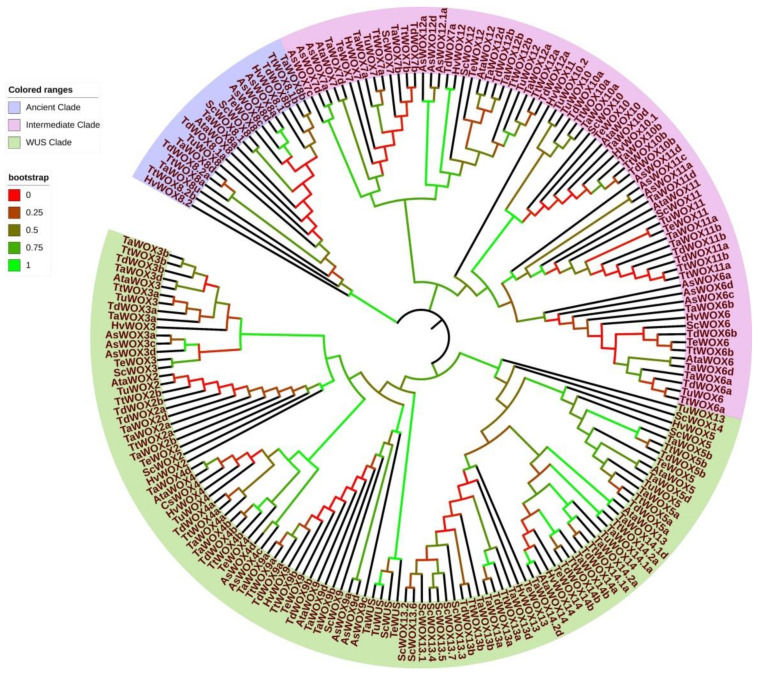
Phylogenetic relationships among WOX proteins from *T. aestivum*, *H. vulgare*, *T. elongatum**, A. sativa*, *S. cereale*, *T. dicoccoides, T. turgidum, A. tauschii*, and *T. urartu.* Phylogenetic tree was constructed based on the sequences of WOX proteins in six Triticeae species implemented by the MEGA X software using maximum likelihood method. Legend in upper left displayed colored ranges of WOX members.

**Figure 6 ijms-22-09325-f006:**
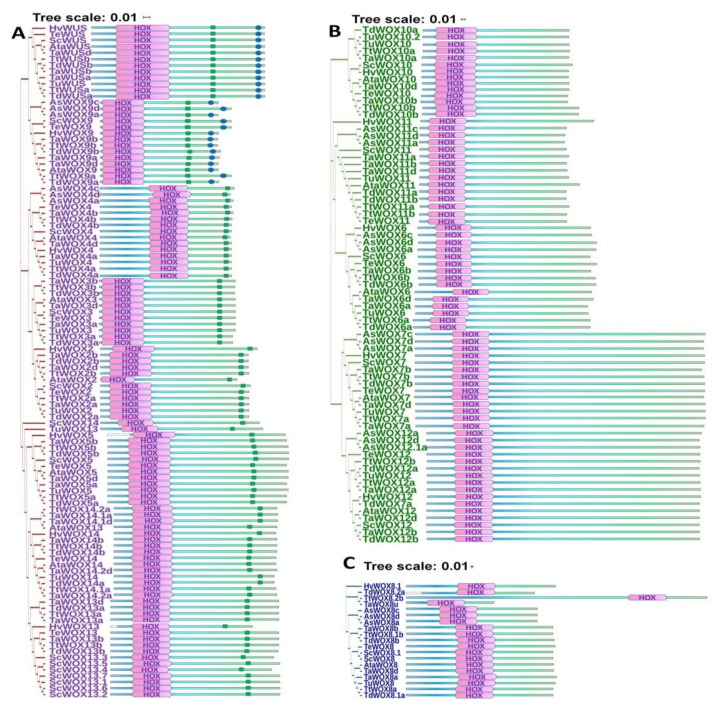
Depiction of the domain structure of WOX proteins in nine Triticeae species. Positions of HOX homeodomain, WUS-Box motif and EAR domain in WOX protein of nine Triticeae species including *T. aestivum*, *H. vulgare*, *T. elongatum**, A. sativa*, *S. cereale*, *T. dicoccoides, T. turgidum, A. tauschii*, and *T. urartu*. WOX members were divided by their phylogenetic relationship. There were HOX homeodomain, WUS-Box motif and EAR domain in WUS clade WOX proteins in these nine Triticeae species (**A**). Positions of HOX homeodomain in intermediate clade WOX proteins in these nine Triticeae species (**B**). Positions of HOX homeodomain in ancient clade WOX proteins in these nine Triticeae species (**C**). Length of WOX members were displayed by the bar length, positions of HOX homeodomain were displayed as pink hexagon, WUS-Box motif was displayed as green round dot, and EAR domain was displayed as blue round dot.

**Figure 7 ijms-22-09325-f007:**
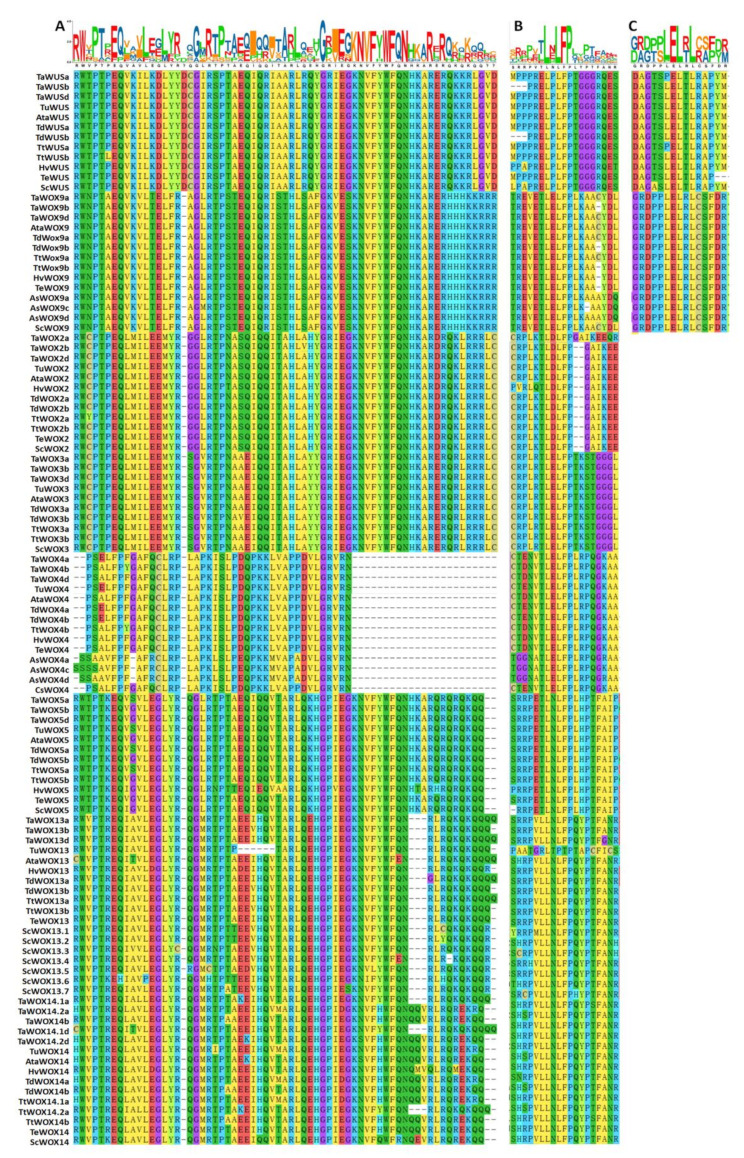
Alignment of WOX homeodomains from WUS clade of nine Triticeae species. Phylogenetic alignment of homeodomain sequences was conducted by ClustalW algorithm using MEGA X software. LOGO of protein sequences represent the relative frequency of an amino acid at the corresponding position, and the content of the aligned sequences at a position in bit (max. 4.322 bit for proteins, i.e., log220). Multiple sequence alignment among the HOX homeodomain (**A**), WUS-motif (**B**) and EAR domain (**C**) of WOX proteins in WUS clade in these nine Triticeae species.

**Figure 8 ijms-22-09325-f008:**
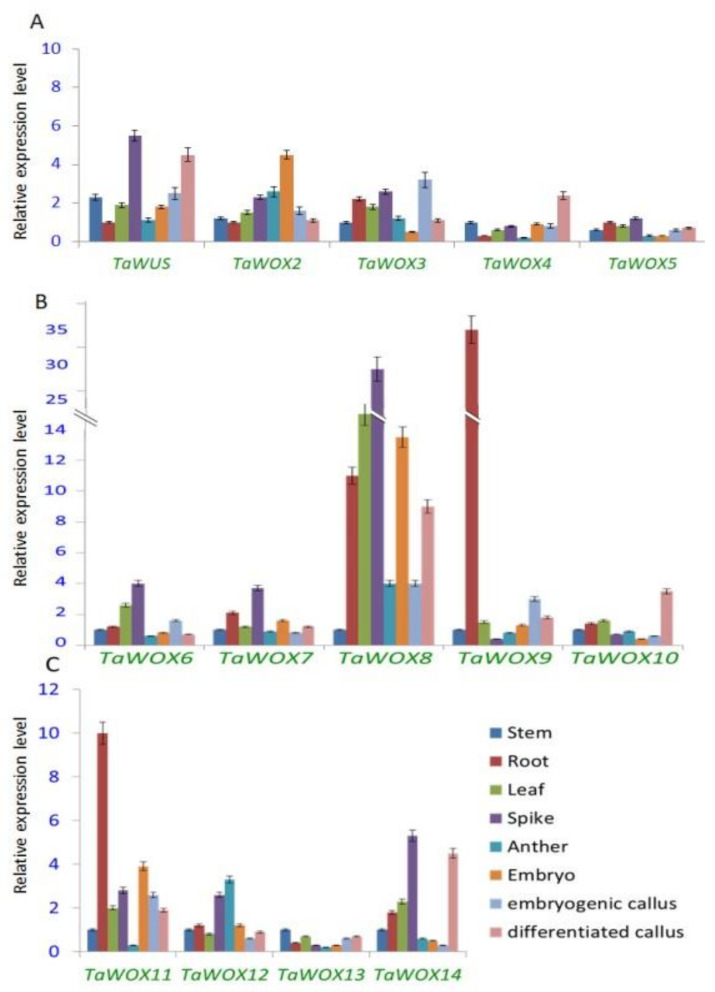
Expression pattern of *TaWOX* genes in various tissues of *T. aestivum*. Gene expression level was examined using qPCR. The qPCR data was normalized using wheat *TaActin* gene. Values were means ± sd of three biological replicates. Expression pattern of *TaWUS*, *TaWOX2*-*TaWOX5* (**A**); expression pattern of *TaWOX6*-*TaWOX10* (**B**); *TaWOX11*-*TaWOX14* (**C**) were shown.

**Figure 9 ijms-22-09325-f009:**
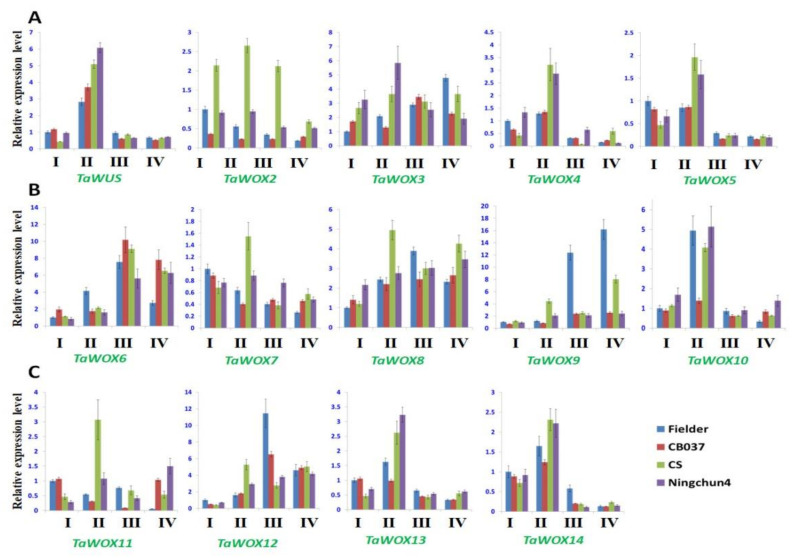
Expression patterns of *TaWOX* genes at different stage in callus proliferation of *T. aestivum*. The expression levels of 14 *TaWOX* genes in the immature embryos derived callus of four wheat cultivars Fielder, CB037, CS, and Ningchun4 at different culture stage were examined using qPCR. The qPCR data was normalized using wheat *TaActin* gene. Values were means ± sd of three biological replicates. I, fresh embryos; II, callus cultured for one week; III, callus cultured for two weeks; IV, callus cultured for three weeks. Expression pattern of *TaWUS*, *TaWOX2*-*TaWOX5* (**A**); expression pattern of *TaWOX6*-*TaWOX10* (**B**); *TaWOX11*-*TaWOX14* (**C**); were shown.

**Table 1 ijms-22-09325-t001:** Characteristics of *TaWOX* gene family members in *T. aestivum*.

Gene	Gene Locus	Chromosome	Gene Stretch Region	mRNA Length (bp)	Protein Sequence Length (aa)	UniProt ID
*TaWOX2a*	*TraesCS1A02G052000*	1A	33,397,501-33,398,955:−1	1314	263	A0A1D5S1T3
*TaWOX12a*	*TraesCS1A02G399400*	1A	563,818,671-563,823,103:1	1854	486	A0A1D5RPD4
*TaWOX2b*	*TraesCS1B02G069000*	1B	53,364,615-53,365,864:−1	1119	264	A0A1B1XWM5
*TaWOX12b*	*TraesCS1B02G427400*	1B	652,781,930-652,786,496:1	1983	485	A0A1D5SDQ8
*TaWOX2d*	*TraesCS1D02G054000*	1D	35,059,826-35,061,088:−1	1138	267	W5ANF9
*TaWOX12d*	*TraesCS1D02G406900*	1D	470,219,711-470,224,514:1	2028	486	A0A1D5SWV6
*TaWUS*	*TraesCS2A02G491900*	2A	724,513,458-724,514,647:1	927	308	A0A1D5TC72
*TaWOX4a*	*TraesCS2A02G514000*	2A	738,371,677-738,372,966:1	1061	234	A0A1D5TF70
*TaWOX11a*	*TraesCS2A02G100700*	2A	53,782,606-53,785,288:1	1380	265	A0A1D5TJV0
*TaWOX11b*	*TraesCS2B02G117900*	2B	81,755,546-81,758,516:1	1366	261	A0A1D5U6K9
*TaWOX4b*	*TraesCS2B02G542600*	2B	740,320,190-740,321,561:−1	1002	237	W5BBK8
*TaWOX11d*	*TraesCS2D02G100200*	2D	52,227,203-52,229,885:1	1379	264	A0A1D5V0E6
*TaWOX4d*	*TraesCS2D02G515600*	2D	606,709,221-606,710,431:1	979	237	A0A1D5UH04
*TaWOX10a*	*TraesCS3A02G073500*	3A	45,776,166-45,777,448:1	992	260	A0A1D5VKG7
*TaWOX7a*	*TraesCS3A02G247200*	3A	465,225,214-465,228,773:1	1968	515	A0A1D5V4S9
*TaWOX8a*	*TraesCS3A02G341700*	3A	588,932,808-588,937,056:1	2230	265	A0A077RTA5
*TaWOX14.1a*	*TraesCS3A02G358200*	3A	606,515,981-606,519,197:−1	1162	288	A0A1D5VFV1
*TaWOX13a*	*TraesCS3A02G358100*	3A	606,444,775-606,446,830:−1	1138	301	A0A1D5VA42
*TaWOX14.2a*	*TraesCS3A02G358400*	3A	606,573,438-606,576,220:−1	1133	290	A0A1D6RQ92
*TaWOX9a*	*TraesCS3A02G368100*	3A	617,060,395-617,061,453:−1	949	212	T1WFN3
*TaWOX10b*	*TraesCS3B02G087800*	3B	56,055,903-56,057,760:−1	1196	261	A0A1D5VWS6
*TaWOX7b*	*TraesCS3B02G272200*	3B	438,378,936-438,382,259:−1	1776	515	A0A077RSZ6
*TaWOX8b*	*TraesCS3B02G373800*	3B	586,694,870-586,698,391:1	1216	261	A0A077S168
*TaWOX13b*	*TraesCS3B02G391100*	3B	616,425,121-616,426,978:−1	900	299	A0A1D5VST7
*TaWOX14b*	*TraesCS3B02G391200*	3B	616,645,332-616,647,892:−1	1216	290	A0A1D5WB93
*TaWOX9b*	*TraesCS3B02G399800*	3B	631,036,656-631,037,718:−1	948	209	D8L9N7
*TaWOX10d*	*TraesCS3D02G073300*	3D	33,294,918-33,295,992:1	786	261	A0A341T564
*TaWOX7d*	*TraesCS3D02G244300*	3D	339,473,290-339,476,679:−1	1834	513	A0A1D5WHW6
*TaWOX8d*	*TraesCS3D02G335500*	3D	447,560,283-447,562,999:1	792	263	A0A341TAX4
*TaWOX13d*	*TraesCS3D02G352500*	3D	463,197,196-463,199,275:−1	1112	298	A0A1D5WMN9
*TaWOX14.1d*	*TraesCS3D02G352600*	3D	463,227,796-463,230,501:−1	895	285	A0A1D5WPP9
*TaWOX14.2d*	*TraesCS3D02G352700*	3D	463,378,560-463,381,808:−1	942	291	A0A1D5WNX7
*TaWOX9d*	*TraesCS3D02G361100*	3D	474,614,857-474,615,873:−1	901	210	T1WGQ3
*TaWOX6a*	*TraesCS4A02G130200*	4A	170,708,103-170,711,065:−1	1350	307	A0A1D5XNI6
*TaWOX6b*	*TraesCS4B02G174400*	4B	382,691,977-382,694,806:1	1254	309	A0A1D5Y4Y9
*TaWOX6d*	*TraesCS4D02G176400*	4D	306,795,298-306,798,208:1	1262	306	A0A341UK30
*TaWOX5a*	*TraesCS5A02G085000*	5A	111,588,730-111,590,895:1	1220	318	A0A341UT17
*TaWOX3a*	*TraesCS5A02G157300*	5A	336,949,988-336,951,183:1	1060	241	A0A1D5YD57
*TaWOX5b*	*TraesCS5B02G091000*	5B	118,451,983-118,454,221:1	1302	321	A0A1D5ZG91
*TaWOX3b*	*TraesCS5B02G156400*	5B	288,891,901-288,893,003:−1	968	241	W5F9A2
*TaWOX5d*	*TraesCS5D02G097400*	5D	108,103,399-108,105,722:1	1381	322	W0Z680
*TaWOX3d*	*TraesCS5D02G162600*	5D	254,023,305-254,024,410:1	1006	242	W5FQU4
*TaWOX8u*	*TraesCSU02G204800*	Un	304,503,012-304,503,827:1	617	156	A0A077RQB3
*TaWUSb*		2B	714,777,526-714,778,733:1	921	306	
*TaWUSd*		2D	590,146,287-590,147,498:1	927	308	
		1D	6,219,571-6,220,231:1			
		3A	64,319,914-64,325,218:−1			
		3B	83,465,544-83,470,232:−1			
		3B	83,471,253-83,471,941:−1			
		3D	52,801,752-52,812,298:−1			
		3D	463,261,309-463,261,744:−1			

**Table 2 ijms-22-09325-t002:** Characteristics of *HvWOX* gene family members in *H. vulgare*.

Gene	Gene Locus	Chromosome	Gene Stretch Region	mRNA Length (bp)	Protein Sequence Length (aa)	Uniprot ID
*HvWOX2*	*HORVU1Hr1G010580*	1H	24,444,001-24,445,742:1	1742	279	A0A287ELV0
*HvWOX12*	*HORVU1Hr1G087940/50*	1H	540,693,806-540,698,431:−1	1470	489	A0A287GM87 A0A287GM65
*HvWOX11*	*HORVU2Hr1G017270*	2H	40,107,707-40,111,565:1	927	308	A0A287H773
*HvWOX4*	*HORVU2Hr1G113820*	2H	729,806,496-729,808,073:1	1151	228	A0A287JHP1
*HvWOX10.1*	*HORVU3Hr1G013290*	3H	28,673,837-28,674,948:−1	786	261	M0Y8G7
*HvWOX10.2*	*HORVU3Hr1G013330*	3H	28,785,048-28,786,156:−1	815	261	A0A287K575
*HvWOX7*	*HORVU3Hr1G060950*	3H	464,417,446-464,421,050:1	2027	516	A0A287L9L2
*HvWOX8.1*	*HORVU3Hr1G080660*	3H	589,829,423-589,834,968:−1	3229	267	M0X0X0
*HvWOX8.2*	*HORVU3Hr1G080690*	3H	590,115,430-590,116,290:1	584	130	A0A287LWD8
*HvWOX9*	*HORVU3Hr1G085050*	3H	610,834,437-610,835,788:−1	1165	209	F2E473
*HvWOX14*	*HORVU3Hr1G086430*	3H	616,993,938-616,996,482:−1	1216	283	M0XTJ6
*HvWOX13*	*HORVU3Hr1G086450*	3H	617,085,484-617,087,698:1	824	274	A0A287M365
*HvWOX6*	*HORVU4Hr1G051530*	4H	423,508,136-423,511,456:−1	1710	306	M0Y4Z0
*HvWOX5*	*HORVU5Hr1G022120*	5H	111,001,136-111,003,388:1	1046	276	A0A287QMF0
*HvWOX3*	*HORVU5Hr1G049190*	5H	381,765,625-381,766,908:1	1126	186	A0A287R4V3
*HvWUS*		2H	717,822,805-717,905,740:−1	942	313	

**Table 3 ijms-22-09325-t003:** Characteristics of *ScWOX* gene family members in *S. cereal*.

Gene	Chromosome	Gene Stretch Region	mRNA Length (bp)	Protein Sequence Length (aa)
*ScWUS*	2R	252,345,136-252,346,331:−1	930	309
*ScWOX2*	1R	48,768,047-48,768,972:−1	789	262
*ScWOX3*	5R	389,070,077-389,070,939:1	726	241
*ScWOX4*	2R	267,063,777-267,064,860:1	705	224
*ScWOX5*	5R	130,892,024-130,893,989:1	927	308
*ScWOX6*	7R	341,262,761-341,264,920:−1	921	306
*ScWOX7*	3R	358,251,618-358,254,632:−1	1545	514
*ScWOX8.1*	3R	102,984,340-102,987,559:1	792	263
*ScWOX8.2*	3R	104,728,944-104,732,163:1	792	263
*ScWOX9*	3R	154,267,694-154,268,435:−1	636	211
*ScWOX10*	3R	107,456,405-107,457,718:−1	786	261
*ScWOX11.1*	2R	77,444,261-77,446,289:1	795	264
*ScWOX11.2*	2R	77,562,117-77,564,149:1	795	264
*ScWOX12*	1R	91,075,823-91,079,684:1	1449	482
*ScWOX13.1*	3R	139,781,793-139,779,696:1	900	299
*ScWOX13.2*	3R	187,095,786-187,097,863:−1	897	298
*ScWOX13.3*	3R	164,219,576-164,221,728:−1	897	290
*ScWOX13.4*	3R	140,032,819-140,034,973:−1	897	286
*ScWOX13.5*	3R	140,194,495-140,196,649:1	897	298
*ScWOX13.6*	3R	157,852,945-157,855,106:1	897	298
*ScWOX13.7*	3R	140,118,810-140,120,961:−1	894	267
*ScWOX13.8*	3R	140138607-140140760:−1	366	121
*ScWOX14*	3R	140,085,774-140,087,928:1	856	283

**Table 4 ijms-22-09325-t004:** Characteristics of *TeWOX* gene family members in *T. elongatum*.

Gene	Chromosome	Gene Stretch Region	mRNA Length (bp)	Protein Sequence Length (aa)
*TeWUS*	2E	636,539,269-636,540,438:1	924	329
*TeWOX2*	1E	62,834,291-62,835,214:−1	789	262
*TeWOX3*	5E	249,414,904-249,415,769:1	732	243
*TeWOX4*	2E	657,284,040-657,284,981:−1	708	235
*TeWOX5*	5E	123,902,262-123,904,308:1	960	319
*TeWOX6*	4E	373,518,174-373,520,519:1	933	310
*TeWOX7*	3E	355,150,216-355,153,543:−1	1512	503
*TeWOX8*	3E	460,009,235-460,013,048:1	789	262
*TeWOX9*	3E	489,665,059-489,665,812:−1	633	210
*TeWOX10*	3E	63,048,238-63,049,186:1	786	261
*TeWOX11*	2E	101,851,313-101,853,580:1	780	259
*TeWOX12*	1E	503,347,861-503,351,437:1	1458	485
*TeWOX13*	3E	480,682,493-480,684,518:−1	903	300
*TeWOX14*	3E	480,874,716-480,877,294:−1	873	290

**Table 5 ijms-22-09325-t005:** Characteristics of *AsWOX* gene family members in *A. sativa*.

Gene	Chromosome	Gene Stretch Region	mRNA Length (bp)	Protein Sequence Length (aa)
*AsWUSa*	2A	Fragments		
*AsWUSc*	2C	Fragments		
*AsWUSd*	2D	Fragments		
*AsWOX2a*	1A	Fragments		
*AsWOX2c*	1C	Fragments		
*AsWOX2d*	1D	Fragments		
*AsWOX3a*	5A	356,256,138-356,257,025:−1	741	246
*AsWOX3c*	5C	395,717,807-395,718,695:−1	738	245
*AsWOX3d*	5D	322,024,845-322,025,729:−1	735	244
*AsWOX4a*	2A	396,219,690-396,220,495:−1	720	239
*AsWOX4c*	2C	549,486,552-549,487,356:−1	717	238
*AsWOX4d*	2D	186,662,767-186,663,482:−1	708	235
*AsWOX5a*	4A	Fragments		
*AsWOX5c*	3C	Fragments		
*AsWOX5d*	3D	Fragments		
*AsWOX6a*	5A	53,538,777-53,540,999:−1	969	322
*AsWOX6c*	4C	594,589,370-594,591,649:1	987	328
*AsWOX6d*	5D	21,782,750-21,784,999:−1	978	325
*AsWOX7a*	3A	376,179,787-376,182,862:−1	1485	494
*AsWOX7c*	3C	514,164,874-514,167,899:−1	1509	502
*AsWOX7d*	3D	327,595,075-327,598,159:−1	1485	494
*AsWOX8a*	4A	369,829,463-369,832,224:−1	741	246
*AsWOX8c*	3C	593,558,934-593,561,981:−1	738	245
*AsWOX8d*	3D	377,117,878-377,121,109:−1	735	244
*AsWOX9a*	4A	401,138,523-401,139,247:−1	633	210
*AsWOX9c*	3C	621024950-621025679:−1	636	211
*AsWOX9d*	4D	357331495-357332237:−1	651	216
*AsWOX10a*	6A	Fragments		
*AsWOX10.1c*	4C	Fragments		
*AsWOX10.2c*	7C	Fragments		
*AsWOX11a*	6A	410997915-410999746:−1	777	258
*AsWOX11c*	4C	24,506,573-24,508,358:1	786	261
*AsWOX11d*	5D	454,073,395-454,075,249:−1	777	258
*AsWOX12a*	1A	373,498,261-373,501,827:1	1416	471
*AsWOX12.1a*	1A	522,889,272-522,892,564:1	1404	467
*AsWOX12d*	1D	357,683,263-357,687,060:1	1413	470

## Data Availability

All data used or analyzed in this study are included in this published article and additional files. Twenty-six predicted WOX family protein sequences of wheat could be downloaded from the Plant TFDB database (http://planttfdb.cbi.pku.edu.cn). The genome sequences and annotation of *WOX* genes in nine Triticeae species could be downloaded from Gramene (http://ensembl.gramene.org/Tools/Blast), URGI (https://urgi.versailles.inra.fr), GrainGenes (https://wheat.pw.usda.gov), and WheatOmics (http://202.194.139.32/). Transcriptome data used for gene expression analysis could be downloaded from expVIP (http://wheat-expression.com/).

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
