# Peer review of "Genome-Wide Identification and Expression Profiling Analysis of WOX Family Protein-Encoded Genes in Triticeae Species"

_ijms, 2021, doi:10.3390/ijms22179325_

Round 1

Reviewer 1 Report

The writing of manuscript was generally acceptable except some choices of words or organization of phrases. The following are examples.

Title – … analysis of genes encoding WOX family proteins in Triticeae species. Throughout the text, “Triticeae plant species” should be shortened to “Triticeae species”

Line 20 – delete “conserved”

Line 32 – Are you referring to “annual Triticeae plants” only in this sentence?

Line 37 – delete “better”

Line 40 – legged behind “that” of rice and maize

Line 44 – “a big limitation” >> “an obstacle”

Line 51 – “extensively commercial varieties” >> “extensively cultivated varieties”

Line 60 - “in molecular level” >> “at molecular level”

Line 68 – indica should be in italic

Line 290 – Analysis “of” the conserved motifs…..

Line 382 – genome assembly and annotation

Line 407 – confusing to some extent

Line 409 – “was” regarded as TaWOX9 due to “its high” similarity to

Line 563 – in “annual” Triticeae species

Author Response

Dear Reviewer,

Thanks a lot for your thoughtful comments on our manuscript entitled “Genome-wide identification and expression profiling analysis of WOX family proteins encoded genes in Triticeae species” (ID: ijms-1285515). We have carefully revised the manuscript according to your comments and suggestions. All the changes are highlighted intrack change model in the revised manuscript. Our point-by-point responses to your comments are presented below.

  1. Title – … analysis of genes encoding WOX family proteins in Triticeae species. Throughout the text, “Triticeae plant species” should be shortened to “Triticeae species”.

Response: Thanks for the kind suggestion. We have edited the phrase in the revised manuscript. In the title as well as the whole manuscript, the statements of “Triticeae plant species” have been changed to “Triticeae species”.

  1. Line 20 – delete “conserved”.

Response: We have deleted the word in the revised manuscript.

  1. Line 32 – Are you referring to “annual Triticeae plants” only in this sentence?

Response: Thanks for the suggestion. We have changed “111 species” to “111 annual species” because these plants mentioned in this sentence are cultivated crops.

  1. Line 37 – delete “better”.

Response: Thanks for the correction. We have deleted “better” here in the revised manuscript.

  1. Line 40 – legged behind “that” of rice and maize

Response: We have revised the phrase by the suggestion in the revised manuscript.

  1. Line 44 – “a big limitation” >> “an obstacle”

Response: Thanks for the comment. We have edited the phrase in the revised manuscript.

  1. Line 51 – “extensively commercial varieties” >> “extensively cultivated varieties”

Response: We have changed “extensively commercial varieties” to “extensively cultivated varieties” in the revised manuscript.

  1. Line 60 - “in molecular level” >> “at molecular level”

Response: We have changed the word in the revised manuscript.

  1. Line 68 – indica should be in italic

Response: Thanks. We have written “indica” in italic.

  1. Line 290 – Analysis “of” the conserved motifs…..

Response: Thanks. We have edited the phrase from “Analysis for the conserved motifs” into “Analysis of the conserved motifs” in the revised manuscript.

  1. Line 382 – genome assembly and annotation.

Response: Thanks, we have edited the phrase from “assemble and annotation of the colossal wheat genome” into “wheat genome assembly and annotation” in the revised manuscript.

  1. Line 407 – confusing to some extent.

Response: We have changed the word in the revised manuscript. We have edited the word from “confused” into “confusing”.

Line 409 – “was” regarded as TaWOX9 due to “its high” similarity to

Response: Thanks a lot for the suggestion. We have edited the phrase from “were regarded as TaWOX9 due to their highly similarity” into “was regarded as TaWOX9 due to its high similarity” in the revised manuscript.

Line 563 – in “annual” Triticeae species.

Response: We have edited the phrase, from “Triticeae species”into “annual Triticeae species” in the revised manuscript.

Thanks again for your time in reviewing our manuscript.

Sincerely yours,

Lei Shi and Xingguo Ye

[email protected], [email protected]

Institute of Crop Sciences

Chinese Academy of Agricultural Sciences

Reviewer 2 Report

WOX-like genes are crucial in plant development and regeneration. This study set out to phylogenetically analyze the WOX-like gene family in several Triticeae species whose genome sequences are publicly available. The authors also determined the chromosome designation for each WOX-like homolog along the chromosome, which was further confirmed using a set of substitution lines. They also examined the expression patterns of these WOX-like genes and found that TaWOX9 might contribute to the regulation of regeneration process in common bread wheat. While the work itself was properly designed and performed with no major issues, I found the conclusions are too weak largely because many of the analyses in this work are confirmative, and no in-depth analyses was conducted to truly advance our understanding the biology of WOX-like genes – otherwise, they might have to convince me the true novelty of this work.

Major thoughts -

  1. A much thorough search for the WOX-like genes is needed for each focal genome, as using the blastp/blastn along against the annotated coding sequences alone is not sufficient to identify all the WOX-like genes – the genomes in wheat/Triticeae in general are notoriously known for its immense size, and thus the difficulty for a complete annotation. There are many genes that are actually not fully annotated. That being said, along with the phylogenetic analyses/results, some further against-genome search would be needed for the conclusion regarding the “loss” of paralogs. In fact, there are just so many information published already on this subject, and this study in fact only include two groups, barley and a set of wheat species. It would make much more sense to include Thinopyrum and Avena genomes that are recently published.
  2. Assigning genes onto the chromosome, this in fact requires only the plotting as far as I understood in this work using the coordinates from the published genomes, although they have confirmed the location experimentally. But I really donot see the “analysis” or “synthesis” here – collinearity in the cereal genome has been known widely and for a long time. Therefore, a synteny analysis would be appreciated, which would greatly assist the presentation of the results and the discussion.

Minor concerns -

There are a number of terms are used improperly – for instance “transcripts”, transcripts are commonly referred to as isoforms of expressed transcripts for a gene, but there are not equivalent of a gene. And I have listed below along with other minor comments.

Line 15, “database of the six Triticeae species”, remove “the”, Triticeae have several hundred species but not six. Also, database to databases.

Line 32-35, this sentence is rather confusing, please rephrase.

Line 39 “up to” or greater than?

Line 40 “legged behind”??

Line 57, “which express”

Line 77, “Update” this sounds weird here.

Line 81, “AtWUS expresses”, check grammar, AtWUS is expressed. This seems to be mis-used across the text.

Line 105-106, “primordials”, “primordial”

Line 109, “closely homologs” closely-related, or sister homologs

Line 142, the word “transcripts” in many places of the manuscript is misused for genes.

Line 165, “AtaWUS, TaWUSb, TaWUSd, TdWUSb, and TtWUSb” italic, cis-regulatory regions are found in regulatory regions of genes, not protein.

Line 175, “alleles”, be careful with the terms, homeologs, alleles, homologs, orthologs, paralogs.

Author Response

Dear Reviewer,

We would first like to express our thanks for handling our submission and for reviewing our manuscript entitled “Genome-wide identification and expression profiling analysis of WOX family proteins encoded genes in Triticeae species” (ID: ijms-1285515). We have now carefully revised the manuscript according to the helpful guidance about how to improve our study. Our full point-by-point responses are provided below.

  1. A much thorough search for the WOX-like genes is needed for each focal genome, as using the blastp/blastn along against the annotated coding sequences alone is not sufficient to identify all the WOX-like genes – the genomes in wheat/Triticeae in general are notoriously known for its immense size, and thus the difficulty for a complete annotation. There are many genes that are actually not fully annotated. That being said, along with the phylogenetic analyses/results, some further against-genome search would be needed for the conclusion regarding the “loss” of paralogs. In fact, there are just so many information published already on this subject, and this study in fact only include two groups, barley and a set of wheat species. It would make much more sense to include Thinopyrum and Avena genomes that are recently published.

Response: Thanks for the nice comments and suggestions. We have made efforts to identify all the WOX-like genes by BLAST against all the genomes of the six Triticeae species, not only against the annotated coding sequences. By your suggestion, we have also identified the putative WOX family genes in the sequenced genomes of Thinopyrum elongatum and Avena sativa as well as Secale cereale which were published recently, and have added the new results in the revised manuscript.

  1. Assigning genes onto the chromosome, this in fact requires only the plotting as far as I understood in this work using the coordinates from the published genomes, although they have confirmed the location experimentally. But I really donot see the “analysis” or “synthesis” here – collinearity in the cereal genome has been known widely and for a long time. Therefore, a synteny analysis would be appreciated, which would greatly assist the presentation of the results and the discussion.

Response: Thanks for the good comments. In this study we found that each WOX or WUS homoeologous gene was collinearly located on the corresponding chromosome among the nine Triticeae species. For examples, WOX2 and 12 were located on chromosome group 1 in T. aestivum, H. vulgare, S. cereale, T. elongatum, and A. sativa; WOX4 and WUS were located on chromosome group 2 in the five species; WOX9 was located on chromosome group 3 in the five species; WOX3 was located on chromosome group 5 in the five species. Our results indicated that the WOX or WUS homoeologous genes in Triticeae species were originated via orthologous evolution approach. We have added these analysis description in the revised manuscript, and the detailed results on collinear chromosome location of the identified WOX genes can be clearly seen in Figure 2 and Tables 1-5.

  1. There are a number of terms are used improperly – for instance “transcripts”, transcripts are commonly referred to as isoforms of expressed transcripts for a gene, but there are not equivalent of a gene. And I have listed below along with other minor comments.

Response: Thanks for the critical comments. We have changed “transcripts” to “putative genes” or “coding region of WOX genes” in the revised manuscript throughout the text. We have also corrected some improperly terms throughout the text.

Line 15, “database of the six Triticeae species”, remove “the”, Triticeae have several hundred species but not six. Also, database to databases.

Response: We have removed “the” in the phrase, and changed “database” to “databases” in the revised manuscript.

Line 32-35, this sentence is rather confusing, please rephrase.

Response: Thanks for the suggestion. The sentence has been rephrased.

Line 39 “up to” or greater than?

Response: We have revised “up to” to “greater than” here by your suggestion.

Line 40 “legged behind”??

Response: We are very sorry for the mistake. We have changed “legged behind” to “lagged behind” in the revised manuscript.

Line 57, “which express”.

Response: We have edited the phrase from “which express” into “expressed” in the revised manuscript.

Line 77, “Update” this sounds weird here.

Response: We have changed “Update published data” to “Published data” in the revised manuscript.

Line 81, “AtWUS expresses”, check grammar, AtWUS is expressed. This seems to be mis-used across the text.

Response: Sorry for the wrong use. We have changed “AtWUS expresses” to “AtWUS expressed” across text in the revised manuscript.

Line 105-106, “primordials”, “primordial”.

Response: Thanks for the correction. We have changed “primordials” to “primordial” in the revised manuscript.

Line 109, “closely homologs” closely-related, or sister homologs

Response: We have changed “closely homologs” to “sister homologs” in the revised manuscript by the suggestion.

Line 142, the word “transcripts” in many places of the manuscript is misused for genes.

Response: Thanks for the comment. The improperly used “transcripts” has been changed to “putative genes” or “coding region of WOX genes” throughout the text.

Line 165, “AtaWUS, TaWUSb, TaWUSd, TdWUSb, and TtWUSb” italic, cis-regulatory regions are found in regulatory regions of genes, not protein.

Response: Thanks for the corrections. We have changed the writings of the gene names in italic.

Line 175, “alleles”, be careful with the terms, homeologs, alleles, homologs, orthologs, paralogs.

Response: Thanks for the comment. We have carefully checked the use of the terms “homeologs, alleles, homologs, orthologs, and paralogs” for right meaning in the revised manuscript.

Thank you very much for your investment of your time in the evaluation of our manuscript and for the helpful guidance about how to improve our study.

Sincerely yours,

Lei Shi and Xingguo Ye

[email protected], [email protected]

Institute of Crop Sciences

Chinese Academy of Agricultural Sciences